# New Era in Systemic Sclerosis Treatment: Recently Approved Therapeutics

**DOI:** 10.3390/jcm11154631

**Published:** 2022-08-08

**Authors:** Satoshi Ebata, Asako Yoshizaki-Ogawa, Shinichi Sato, Ayumi Yoshizaki

**Affiliations:** Department of Dermatology, Graduate School of Medicine, The University of Tokyo, Tokyo 113-8655, Japan

**Keywords:** systemic sclerosis, interstitial lung disease, nintedanib, tocilizumab, rituximab, SENSCIS study, FocuSSced study, DESIRES study, modified Rodnan skin score, forced vital capacity

## Abstract

Systemic sclerosis (SSc) is a chronic autoimmune disease with a poor prognosis. Among the various complications of SSc, treatment options for the fibrotic lesions, skin sclerosis, and SSc-associated interstitial lung disease (SSc-ILD) have been limited. However, since 2019, the efficacy and safety of nintedanib, tocilizumab, and rituximab for SSc or SSc-ILD have been demonstrated in double-blind, randomized, placebo-controlled trials, respectively. The antifibrotic agent nintedanib was approved for SSc-ILD in all regions of the United States, Europe, and Japan after the SENSCIS study confirmed that it suppressed the reduction in forced vital capacity (FVC), a measure of SSc-ILD. Tocilizumab, an anti-interleukin-6 receptor antibody, was approved for the treatment of SSc-ILD in the United States after the FocuSSced study showed that it inhibited the decrease in FVC. Rituximab, an anti-CD20 antibody, showed improvement in both modified Rodnan skin score, a measure of skin sclerosis, and FVC in the DESIRES study, and was approved in Japan for the treatment of SSc itself. With the development of these three drugs, SSc treatment is entering a new era. This paper outlines the latest advances in SSc therapeutics, focusing on nintedanib, tocilizumab, and rituximab.

## 1. Introduction

Systemic sclerosis (SSc) is a connective tissue disease caused by abnormal activation of the immune system [1,2,3]. Characterized by vascular damage and fibrosis in various organs, the disease has a high frequency of complications, a poor prognosis, and high unmet medical needs [4,5,6,7,8].

Among the complications of SSc, potent therapeutic agents have already been found for the vascular complications of renal crisis and SSc-associated pulmonary arterial hypertension (SSc-PAH). Advances in treatment algorithms have resulted in lower mortality rates from these complications than in the past [9,10,11,12,13,14,15,16,17,18]. For renal crisis, treatment of hypertension with angiotensin-converting enzyme inhibitors has markedly improved prognosis [19]. SSc-PAH remains a difficult complication to manage, but treatment with pulmonary vasodilators early in the course of the disease has improved outcomes. A sub-analysis of the AMBITION study revealed that the combination of ambrisentan, an endothelin receptor antagonist, and tadalafil, a phosphodiesterase type 5 inhibitor, was greatly effective in the treatment of SSc-PAH [20].

In contrast, the development of sufficiently effective and safe therapies for the fibrosis symptoms caused by SSc, such as skin sclerosis and SSc-associated interstitial lung disease (SSc-ILD), has been slow and remains a major challenge [18,21,22,23]. Previously, cyclophosphamide was the only drug shown to be effective for skin sclerosis or ILD in SSc patients in a double-blind, randomized, or placebo-controlled trial [24]. However, cyclophosphamide is carcinogenic and cannot be administered long-term. The therapeutic effects of cyclophosphamide on SSc do not last for 2 years [25]. Other than cyclophosphamide, immunosuppressive therapy with methotrexate or mycophenolate mofetil may be used for fibrotic lesions in SSc. However, methotrexate failed to show significant improvements in skin sclerosis compared with the placebo in previous randomized controlled trials [26,27]. Mycophenolate mofetil was reported in the SLS II study to be not clearly different from cyclophosphamide in improving FVC for patients with SSc-ILD. However, mycophenolate mofetil failed to show superiority over cyclophosphamide in this study [28]. Additionally, the efficacy of mycophenolate mofetil has not been validated in a placebo-controlled randomized controlled trial. For these reasons, neither methotrexate nor mycophenolate mofetil is approved for SSc and is off-label when used. Autologous hematopoietic stem cell transplantation (HSCT) has been reported to significantly improve skin sclerosis and SSc-ILD compared to cyclophosphamide in open-label, randomized, and controlled trials [29,30,31,32]. However, a major problem with autologous HSCT is the high rate of treatment-related adverse events and mortality. Safety issues require careful consideration of the indications for autologous HSCT and limit the patients for whom it should be indicated [33].

Thus, treatment for skin sclerosis and ILD, important complications of SSc, has not been well established. Skin sclerosis is the most important indicator in the diagnostic criteria for SSc [34]. Its degree has been reported to correlate with poor prognosis in SSc patients [35]. In addition, SSc-ILD is the most common direct cause of death in SSc, surpassing renal crisis and SSc-PAH [4,36]. The development of therapies for fibrotic lesions in SSc is highly desirable.

In this context, a series of double-blind, randomized, and placebo-controlled trials of nintedanib, tocilizumab, and rituximab have published results after 2019. Since September 2019, the United States Food and Drug Administration (FDA), the Japanese Ministry of Health, Labor, and Welfare (MHLW), and the European Medicines Agency (EMA) have successively approved nintedanib for SSc-ILD based on the results of the SENSCIS study [37]. Then, in March 2021, following the results of the FocuSSced study [38], the FDA approved tocilizumab, also for SSc-ILD. Subsequently, in September 2021, the Japanese MHLW approved rituximab for SSc on the basis of the results of the DESIRES study [39]. All of these therapies are expected to inhibit or improve tissue fibrosis caused by SSc. Table 1 summarizes the approval status of each therapeutic agent in the United States, Europe, and Japan.

The emergence of nintedanib, tocilizumab, and rituximab has greatly expanded the treatment options for skin sclerosis and SSc-ILD, opening a new era of SSc treatment. This paper provides an overview of these three drugs approved in major countries, focusing on the results of clinical trials that formed the basis for their approval.

## 2. Nintedanib

Nintedanib, an indolinone derivative, is a triple kinase inhibitor that strongly inhibits vascular endothelial growth factor receptor, fibroblast growth factor receptor, and platelet-derived growth factor receptor [40]. It suppresses fibroblast proliferation by blocking mitogen-activated protein kinase and Akt signaling pathways in three types of cells involved in angiogenesis: endothelial cells, pericytes, and smooth muscle cells [41].

Nintedanib was first clinically applied for idiopathic pulmonary fibrosis (IPF). In the INPULSIS studies, nintedanib was shown to significantly suppress the annual rate of decrease in forced vital capacity (FVC) in patients with IPF compared with the placebo [42]. The adjusted annual rate of change in FVC was −114.7 mL in the nintedanib group and −239.9 mL in the placebo group in the INPULSIS-1 study (difference 125.3 [95% CI 77.7 to 172.8]; *p* < 0.001), and −113.6 mL in the nintedanib group and −207.3 mL in the placebo group in the INPULSIS-2 study (difference 93.7 [95% CI 44.8 to 142.7]; *p* < 0.001). This means that while nintedanib did not improve IPF, it suppressed the degree of FVC reduction by approximately half. Approved by the FDA in October 2014 for the treatment of IPF, nintedanib is recommended for conditional use in the IPF treatment guidelines of the American Thoracic Society, the European Respiratory Society, the Japanese Respiratory Society, and the Latin American Thoracic Association [43].

Having demonstrated antifibrotic effects in patients with IPF, nintedanib was subsequently investigated to expand its application in SSc, a prototype of systemic fibrosis. From 2015 to 2018, the SENSCIS study was conducted on 576 SSc-ILD patients from 32 countries [37]. Patients were randomized 1:1 to orally receive 150 mg of nintedanib twice daily or the placebo and were evaluated 52 weeks after initiation of the study drug. The adjusted annual change in FVC was −52.4 mL/year in the nintedanib group and −93.3 mL/year in the placebo group (difference 41.0 [95% CI 2.9 to 79.0]; *p* = 0.04). As with IPF, nintedanib did not improve or maintain FVC, but significantly suppressed the degree of FVC reduction. Based on the results of this study, nintedanib for SSc-ILD is now covered by insurance in many regions, with approval from the FDA in September 2019, the Japanese MHLW in December 2019, and the EMA in April 2020.

A unique aspect of the SENSCIS study was that the study drug was allowed to be administered in combination with immunosuppressive agents, such as mycophenolate mofetil. Post-hoc analyses showed that the mean adjusted annual decrease in FVC for patients taking mycophenolate mofetil at baseline was −40.2 mL in the nintedanib group and −66.5 mL in the placebo group (difference 26.3 [95% CI −27.9 to 80.6]). For patients not taking mycophenolate mofetil at baseline, the mean adjusted annual decrease in FVC was −63.9 mL in the nintedanib group and −119.3 mL in the placebo group (difference 55.4 [95% CI 2.3 to 108.5]) [44]. Nintedanib, with or without mycophenolate mofetil, demonstrated better treatment results than the placebo. When nintedanib was used in combination with mycophenolate mofetil, the degree of inhibition of FVC reduction was greater than when it was used alone. Nintedanib, an antifibrotic agent, has a different mechanism of action than the immunosuppressive agents that have been the standard treatment for SSc-ILD. It has been suggested that the combination of an antifibrotic and an immunosuppressive agent may increase the therapeutic efficacy against lung fibrosis in SSc patients.

In contrast, nintedanib does not appear to be very effective for fibrotic lesions in the skin. The change from baseline in the modified Rodnan skin score (mRSS) at 52 weeks, a secondary endpoint of the SENSCIS study, was −2.17 in the nintedanib group and −1.96 in the placebo group, with no statistically significant difference (difference −0.21 [95% CI −0.94 to 0.53]; *p* = 0.58) [37]. A sub-analysis differentiated by the presence or absence of mycophenolate mofetil was also performed, but again, there was no significant difference in mRSS improvements between the nintedanib and the placebo groups in either case. For skin fibrosis, there may be little benefit to adding an antifibrotic agent to immunosuppressive drugs. The disease for which nintedanib was approved is SSc-ILD, not SSc itself. If SSc patients with severe skin sclerosis are to be treated, the therapeutic strategy should focus on agents other than nintedanib.

It should be noted that gastrointestinal side effects are common with nintedanib. In the SENSCIS study, 75.7% of nintedanib-treated patients had diarrhea, 31.6% had nausea, 24.7% had vomiting, and 11.8% had weight loss. The incidence of all of these was more than twice that of the placebo-treated patients [37]. Perhaps because of the higher incidence of adverse events, total scores on the St George’s Respiratory Questionnaire, a measure of health-related quality of life, were worse in the nintedanib group than in the placebo group. Specifically, in a sub-analysis of patients younger than 65 years only, the change from baseline in the total score on the St George’s Respiratory Questionnaire was significantly worse in the nintedanib group than in the placebo group (difference 2.8 [95% CI 0.1 to 5.6]). How to manage the incidence of adverse events and deterioration in the quality of life caused by nintedanib remains a challenge. A study in patients with IPF suggested that reducing the dose of nintedanib or combining it with multiple antidiarrheal agents, such as clostridium butyricum, may be effective in the management of diarrhea [45]. A similar approach may be necessary when using nintedanib in SSc-ILD patients.

A recent topic outside of SSc-ILD is the FDA’s approval of nintedanib for progressive fibrosing ILD in March 2020 based on the INBUILD study [46], expanding the indication for nintedanib. In the IMBUILD trial, as in the INPULSIS and SENSCIS studies, FVC reduction was suppressed more in the nintedanib group than in the placebo group. The results presented in these trials are similar, suggesting that nintedanib may be effective in preventing pulmonary fibrosis independent of the disease type. The availability of nintedanib for multiple diseases increases the number of patients for whom nintedanib is indicated. Regardless of SSc-ILD, it is hoped that more real-world evidence for nintedanib will be accumulated in the future.

## 3. Tocilizumab

Tocilizumab is a recombinant humanized antihuman interleukin 6 (IL-6) receptor monoclonal antibody that binds to the IL-6 receptor with high affinity [47]. IL-6 is a known inflammatory cytokine. For example, in rheumatoid arthritis (RA), IL-6 is produced locally in the joints, causing joint swelling and elevated C-reactive protein. Tocilizumab suppresses inflammation by inhibiting downstream signaling of IL-6 and is effective against various inflammatory diseases [47,48]. The drug is approved in many countries for RA, Castleman’s disease and juvenile idiopathic arthritis, and is also expected to have therapeutic effects on giant cell arteritis, polymyalgia rheumatica, and large-vessel vasculitis [48]. Tocilizumab was also approved for severe COVID-19 treatment by the EMA in 2021 [49].

IL-6 is thought to be involved in the pathogenesis of SSc. It has been reported that serum IL-6 levels in SSc patients correlate with mRSS [50], and that peripheral blood mononuclear cells of SSc patients produce more IL-6 than those of healthy controls [51]. Among SSc patients, the elevation of serum IL-6 levels is particularly prominent in diffuse cutaneous SSc patients with an early onset of the disease, suggesting a strong influence of IL-6 in the disease progression of SSc [52].

Based on these findings, two double-blind, randomized, and placebo-controlled trials of tocilizumab for SSc were conducted. First, the FaSScinate trial was carried out between 2012 and 2015 in 87 patients with SSc in five countries (Canada, France, Germany, the United Kingdom, and the United States) [53]. Patients were randomized 1:1 to receive 162 mg of tocilizumab or the placebo subcutaneously every week. The primary endpoint of the study was the change from baseline in mRSS after 24 weeks of the study drug administration. The mRSS improved by 3.92 in the tocilizumab group and 1.22 in the placebo group, with the difference between the two groups not being statistically significant (difference −2.70 [95% CI −5.85 to 0.45]; *p* = 0.09). Subsequently, from 2015 to 2019, the FocuSSced trial was conducted on 210 SSc patients from 20 countries in Europe, North America, Latin America, and Japan [38]. As in the FaSScinate trial, patients were randomized 1:1 to receive weekly subcutaneous 162 mg of tocilizumab or the placebo. The primary endpoint of the FocuSSced study was the change in mRSS from baseline at 48 weeks after the study drug administration. The change in mRSS was −6.14 for the tocilizumab group and −4.41 for the placebo group, with no significant difference between the two groups (difference −1.73 [95% CI −3.78–0.32]; *p* = 0.10). In both the FaSScinate and FocuSSced studies, there was a greater improvement in mRSS in the tocilizumab group than in the placebo group, but the difference did not reach statistical significance. The effect of tocilizumab on improving skin fibrosis in SSc patients seems promising, but it has not yet been fully validated and further studies are needed.

In contrast, these two trials suggest that tocilizumab may be effective in patients with SSc-ILD. According to a post-hoc exploratory analysis of the FaSScinate trial, the change in FVC from baseline after 24 weeks of treatment was −34 mL in the tocilizumab group and −171 mL in the placebo group [53]. There was significantly less deterioration in FVC in the tocilizumab group compared with the placebo group (difference 136 [95% CI 9 to 264]; *p* = 0.04). However, the change in FVC from baseline after 48 weeks of treatment was −117 mL in the tocilizumab group and −237 mL in the placebo group, with no statistically significant difference between the two groups (difference 120 [95% CI 23 to 262]; *p* = 0.10). In the FocuSSced study, one of the secondary endpoints was the change from baseline in percent predicted FVC (ppFVC) at 48 weeks after the study drug administration [38]. The placebo group showed a 4.6% decrease in ppFVC, while the tocilizumab group showed only a 0.4% decrease in ppFVC. There was significantly less worsening of ppFVC in the tocilizumab group compared to the placebo group (difference 4.2 [95% CI 2.0 to 6.4]; *p* = 0.0002).

Although data from different trials cannot be simply matched, after 24 weeks of the study drug administration, nintedanib in the SENSCIS trial prevented a 44% decrease in FVC compared to the placebo [37], and tocilizumab in the FocuSSced trial prevented an 87% decrease in FVC compared to the placebo [38]. Tocilizumab may be a potent preventer of SSc-ILD progression. Based on the results of the FocuSSced trial, in March 2021, the FDA approved tocilizumab for SSc-ILD, not SSc. However, neither the FaSScinate nor the FocuSSced trials met their primary endpoints [38,53], and tocilizumab is not currently approved for SSc-ILD in Europe or Japan. To more accurately assess the efficacy of tocilizumab in patients with SSc-ILD, it would be desirable to conduct a clinical trial with an ILD-related measure as the primary endpoint.

The safety of tocilizumab for SSc has been reported to be favorable. The three most frequently reported serious adverse events in the tocilizumab group during the 48-week double-blind period in the FocuSSced study were infections and infestations, cardiac disorders, and skin and subcutaneous tissue disorders. The incidence of all these events was rather higher in the placebo group than in the tocilizumab group [38]. The FocuSSced study included a 48-week open-label period followed by a 48-week double-blind period. During the open-label extension, all patients received tocilizumab, and no major changes in the safety profile were reported [54]. Tocilizumab may be relatively safe to use for at least the two-year treatment period.

## 4. Rituximab

Rituximab is a chimeric antibody against CD20, a cell membrane molecule specifically expressed in B cells. It affects the calcium ion regulatory function of CD20, inhibiting B cell signaling and the cell cycle, thereby eliminating B cells. Antibody-dependent cell-mediated cytotoxic effects and complement-dependent cytotoxic effects are also thought to be involved in the elimination of B cells by rituximab [55].

Rituximab was originally used to treat B-cell lymphomas. In recent years, its efficacy against various auto-inflammatory diseases caused by B cells, such as RA, microscopic polyangiitis, and granulomatosis with polyangiitis, has been successively validated [56]. Although the pathogenesis of SSc remains unclear, B cells are thought to play a central role [57,58]. Abnormal B cell function has been identified in SSc, with increased expression of CD19, which is specifically expressed in B cells [59,60]. Therefore, B-cell removal therapy with rituximab was anticipated to be beneficial in the treatment of SSc.

From 2017 to 2019, the DESIRES study was conducted in Japan with 56 SSc patients [39]. Participants were randomized 1:1 to receive either rituximab 375 mg/m^2^ or the placebo intravenously for four consecutive weeks. The primary endpoint was the change in mRSS from baseline after 24 weeks of the study drug administration. The placebo group showed a 2.14 worsening of mRSS while the rituximab group improved by 6.30, indicating that mRSS was significantly improved in the rituximab group compared to the placebo group (difference −8.44 [95% CI −11.00 to 5.88]; *p* < 0.00001). Rituximab, in the DESIRES study, is the first SSc therapeutic agent to demonstrate efficacy in a double-blind, randomized, or placebo-controlled trial with a primary endpoint of a measure of skin sclerosis. Based on the results, rituximab was approved for SSc in Japan in September 2021. However, the DESIRES study included only Japanese patients, and rituximab has not been approved for SSc in the United States or Europe.

Not only the DESIRES trial, but also meta-analyses integrating clinical trials containing non-Asian patients have reported improvement in mRSS with rituximab treatment [61,62,63]. It is hoped that the efficacy of rituximab in fibrotic involvement of SSc will be tested in future double-blind, placebo-controlled, and randomized controlled trials that include SSc patients from multiple ethnic groups.

After the DESIRES study was completed, a post-hoc analysis was conducted to identify patients who would benefit more from improvements in mRSS with rituximab [64]. After machine learning analysis of 27 candidate predictors, three factors were selected as determinants of the rituximab effect on mRSS improvements: baseline “peripheral blood B-cell count (≥57 µL or <57 µL)”, “mRSS (≥17 or <17)”, and “serum surfactant protein D (SP-D) level (≥151 ng/mL or <151 ng/mL)”. The superiority of rituximab compared with the placebo was not confirmed by SSc patients with low B-cell counts. In contrast, SSc patients with high B-cell counts and high mRSS had greater improvements in mRSS with the rituximab treatment. In patients with high B-cell counts and low mRSS, serum SP-D levels affected the therapeutic effect. B cells are direct therapeutic targets with rituximab, and mRSS and serum SP-D levels are indicators reflecting the degree of skin sclerosis and SSc-ILD, respectively. B-cell removal therapy with rituximab appears to be particularly effective for SSc patients with increased B-cell counts and severe ongoing fibrosis in the skin or lungs. For SSc patients who meet these factors, it may be advisable to positively consider the use of rituximab.

In the DESIRES study, as a secondary endpoint related to SSc-ILD, the change in ppFVC from baseline after 24 weeks of the study drug administration was evaluated [39]. The placebo group had a 2.87% worsening of ppFVC while the rituximab group had a 0.09% improvement in ppFVC, with the rituximab group having significantly better ppFVC than the placebo group (difference 2.96 [95% CI 0.08–5.84]; *p* < 0.04). It is noteworthy that in the DESIRES study, unlike the SENSCIS, FaSScinate, and FocuSSced studies, FVC did not worsen in the actual drug group [37,38,39,53]. In addition, a 24-week open-label extension study in which all patients received rituximab after the double-blind phase of the DESIRES study demonstrated sustained improvements in ppFVC [65]. Rituximab was suggested to be effective in improving SSc-ILD.

The DESIRES trial also confirmed that rituximab is safe for SSc patients. During the 24-week double-blind phase, there was no significant increase in adverse events, including infections, in the rituximab group compared with the placebo group [39]. A subsequent 24-week open-label extension study did not change the safety profile, suggesting that treatment with rituximab is relatively safe for at least two courses and up to 48 weeks [65].

Rituximab is anticipated to be effective for complications besides fibrotic lesions of SSc, and is currently being tested in clinical trials outside of the DESIRES study. The RESTORE sub-study, a randomized-controlled trial, examined the effect of rituximab on SSc-PAH [66]. Although no statistically significant differences were observed in this study, there was a trend toward a more favorable change in 6-min walking distance in the rituximab group compared with the placebo group, suggesting that rituximab is a promising treatment for SSc-PAH [67]. Rituximab has also been reported to be effective against SSc-related polyarthritis in retrospective studies [68]. If the efficacy of rituximab in these complications is also validated, rituximab will considerably change the treatment strategy for SSc.

However, there are many unknowns about the safety of rituximab in the COVID-19 era. The DESIRES trial ended in 2019 [39,65], so the risk of COVID-19 has not been assessed. There is concern that B-cell elimination therapy with rituximab may worsen the prognosis of COVID-19 [69,70,71,72,73,74]. In SSc patients requiring rituximab, it is more important to take thorough infection control measures, such as hand washing and wearing masks [74]. In addition, the administration of rituximab may decrease the efficacy of the COVID-19 vaccine [72,73,74,75,76,77,78,79,80,81]. Therefore, it is preferable to administer the COVID-19 vaccine before the first dose of rituximab [74]. The timing of the COVID-19 vaccination and rituximab treatment should be judged comprehensively based on the prevalence of COVID-19 and the disease status of each SSc patient. Until the COVID-19 era settles down, the appropriateness of the RTX introduction should be carefully determined.

## 5. Conclusions

Therapeutic agents for SSc complications, especially the fibrotic lesions, such as skin sclerosis and ILD, have been limited to date. However, since 2019, based on the results of the SENSCIS, FocuSSced, and DESIRES studies [37,38,39], nintedanib, tocilizumab, and rituximab have been approved in some major countries for SSc or SSc-ILD, respectively.

The results of these three trials differ markedly with respect to their effectiveness in improving skin sclerosis. Table 2 summarizes the changes in mRSS among the outcomes of each trial.

Rituximab is the only drug that showed statistically significant improvements in mRSS in the actual drug group compared with the placebo group [39]. However, because the DESIRES trial was conducted only on Japanese patients, rituximab is approved for SSc only in Japan and not in the United States or Europe. Tocilizumab has been shown to tend to improve mRSS more than the placebo, but this effect has never been shown to be statistically significant. The efficacy of tocilizumab in improving skin fibrosis has not yet been fully validated, and currently, tocilizumab is not approved in any country for SSc itself [38]. Nintedanib has been found to be less effective in improving mRSS [37], and no country has approved SSc itself as well as tocilizumab. It may be difficult to treat SSc patients with severe skin fibrosis with nintedanib alone.

For SSC-ILD, the actual drug group was superior to the placebo group in all of the SENSCIS, FocuSSced, and DESIRES trials. However, there appears to be a difference in treatment efficacy. Table 3 summarizes the outcomes related to FVC from the results of each trial.

Nintedanib has been shown to inhibit FVC reduction and is approved for the treatment of SSc-ILD in all of the United States, Europe, and Japan [37]. However, the efficacy of nintedanib for SSc-ILD is relatively modest. Compared to the placebo, treatment with nintedanib prevents about half of the FVC exacerbations. Nintedanib does not maintain or improve lung function and may be better introduced before SSc-ILD has progressed substantially. Tocilizumab has been reported to significantly prevent FVC decline [38], and its degree of suppression may exceed that of nintedanib [37]. Treatment with tocilizumab may be able to maintain lung function nearly unchanged. However, the efficacy of tocilizumab has never been validated in SSc clinical trials with a significant difference in the primary endpoint. Tocilizumab is approved for SSc-ILD in the United States, but not yet in Europe or Japan. New clinical trials of tocilizumab in SSc-ILD patients may be needed in the future. Rituximab has been suggested to be able to maintain or improve FVC in the DESIRES study of Japanese patients [39,65]. Given that both the nintedanib and tocilizumab trials showed a decrease in FVC from baseline in the actual drug group [37,38], the potential for rituximab to provide improvement in SSc-ILD could make it more appealing than other agents. However, rituximab is approved only in Japan for SSc and has not been fully validated in the United States and Europe. Future double-blind randomized controlled trials are needed to evaluate the efficacy and safety of rituximab in non-Asian patients with SSc.

As mentioned above, nintedanib, tocilizumab, and rituximab have all been reported to be effective in the treatment of fibrotic lesions of SSc but still need to be validated in detail. How to use these agents differently remains to be discussed based on real-world data. Other agents, such as romilukimab and abatacept, are also being studied [82,83], making the competition to develop treatments for SSc more intense. Further research is expected to advance a new era of SSc treatment with agents, such as nintedanib, tocilizumab, and rituximab.

## Figures and Tables

**Table 1 jcm-11-04631-t001:** Approval status of nintedanib, tocilizumab, and rituximab in the United States, Europe, and Japan, respectively, related to systemic sclerosis.

	Indication	United States	Europe	Japan
Nintedanib	SSc-ILD	Approved	Approved	Approved
Tocilizumab	SSc-ILD	Approved	Not Approved	Not Approved
Rituximab	SSc	Not Approved	Not Approved	Approved

SSc = systemic sclerosis. SSc-ILD = systemic sclerosis-associated interstitial lung disease.

**Table 2 jcm-11-04631-t002:** Results of endpoints related to modified Rodnan skin score in each of the studies that formed the basis for approval of nintedanib, tocilizumab, and rituximab for systemic sclerosis or systemic sclerosis-associated interstitial lung disease.

	mRSS-Related Endpoint	Actual Drug Group	Placebo Group	Difference (95% CI)	*p* Value
SENSCIS study [37]	mRSS change from baseline to 52 weeks	−2.17	−1.96	−0.21 (−0.94 to 0.53)	0.58
FocuSSced study [38]	mRSS change from baseline to 48 weeks	−6.1	−4.4	−1.7 (−3.8 to 0.3)	0.10
DESIRES study [39]	mRSS change from baseline to 24 weeks	−6.30	2.14	−8.44 (−11.00 to −5.88)	<0.0001

mRSS = modified Rodnan skin score.

**Table 3 jcm-11-04631-t003:** Results of endpoints related to forced vital capacity in each of the studies that formed the basis for approval of nintedanib, tocilizumab, and rituximab for systemic sclerosis or systemic sclerosis-associated interstitial lung disease.

	FVC-Related Endpoint	Actual Drug Group	Placebo Group	Difference (95% CI)	*p* Value
SENSCIS study [37]	the adjusted annual rate of FVC change	−52.4 mL/year	−93.3 mL/year	41.0 mm/year (2.90 to 79.9)	0.04
FocuSSced study [38]	ppFVC change from baseline to 48 weeks	−0.4%	−4.6%	4.2% (2.0 to 6.4)	0.0002
DESIRES study [39]	ppFVC change from baseline to 24 weeks	0.09%	−2.87%	2.96% (0.08 to 5.84)	0.04

FVC = forced vital capacity. ppFVC = percent predicted forced vital capacity.

## Data Availability

Not applicable.

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
