# Peer review of "New Era in Systemic Sclerosis Treatment: Recently Approved Therapeutics"

_jcm, 2022, doi:10.3390/jcm11154631_

Round 1

Reviewer 1 Report

This is a very well-written and well-organized, concise review of new therapeutics for SSc-ILD/SSc. 

I would suggest removing from the abstract regarding tocilizumab "although it did not verify improvement in modified Rodnan skin score" - an individual only reading abstracts might incorrectly infer that nintedanib and rituximab in contrast were more effective for skin sclerosis without further context. 

Line 332 "the DESIRES trial was conducted only in Japanese" - I would suggest revising to "the DESIRES trial was conducted only in Japanese patients"

Author Response

Comments for the Author

 This is a very well-written and well-organized, concise review of new therapeutics for SSc-ILD/SSc.

Response

 We deeply appreciate your kind words of praise. I also appreciate your careful review of our work. We would like to respond to the points you have pointed out one by one below.

Comment 1

I would suggest removing from the abstract regarding tocilizumab "although it did not verify improvement in modified Rodnan skin score" - an individual only reading abstracts might incorrectly infer that nintedanib and rituximab in contrast were more effective for skin sclerosis without further context.

Response

 Thank you for your important remarks. We agree with you on the points you have raised. We have removed the relevant text as you have indicated.

Comment 2

 Line 332 "the DESIRES trial was conducted only in Japanese" - I would suggest revising to "the DESIRES trial was conducted only in Japanese patients"

Response

 Thank you for your valuable suggestions. We apologize for the confusing description. We have changed "Japanese" to "Japanese Patients" as you recommended.

Reviewer 2 Report

In this article, the authors discuss the complications of systemic sclerosis, including renal and pulmonary complications, and describe the current treatments for the disease and their limitations.

Then, they introduce nintedanib, tocilizumab and rituximab, which have recently been approved at least in one of the three following regions: the USA, Japan and Europe.

They describe the clinical improvements obtained by each drug in the clinical trials that led to their approval.

I have only minor comments:

Tables 2 and 3:

the other clinical trials RESTORE; INBUILD; IMPULSIS; faSScinate are missing

in the table 2, Would it be possible to standardize the parameters assessed for FVC-related endpoint ?

a reference for tolicizumab is missing in relation to its mechanism of action line 169 to 171

line 332 is it japanese that was written or japan? 

Author Response

Comments for the Author

In this article, the authors discuss the complications of systemic sclerosis, including renal and pulmonary complications, and describe the current treatments for the disease and their limitations.

Then, they introduce nintedanib, tocilizumab and rituximab, which have recently been approved at least in one of the three following regions: the USA, Japan and Europe.

They describe the clinical improvements obtained by each drug in the clinical trials that led to their approval.

Response

Thank you very much for your kind review of our paper. Regarding your valuable comments, we will respond to them one by one below.

Comment 1

 I have only minor comments:

Tables 2 and 3:

the other clinical trials RESTORE; INBUILD; IMPULSIS; faSScinate are missing

Response

 Thank you for pointing that out. We apologize that our explanation was not clear. In this Table 2 and Table 3, we only summarize the clinical trials that formed the basis for the approval of each drug for SSc or SSc-ILD. The INBUILD and IMPULSIS trials are for diseases other than SSc. The RESTORE and faSScinate study is an exploratory study and was not the basis for approval for SSc, SSc-ILD, or SSc-PAH. Therefore, they are excluded from Tables 2 and 3.

 In particular, the INBUILD and IMPULSIS trials are not SSc-related trials and therefore do not have mRSS-related endpoints and cannot be included in Table 2. Also, the RESTORE trial is focused solely on PAH and does not report endpoints related to mRSS or FVC, so it cannot be included in either Table 2 or Table 3.

Could we leave the current description as it is? We would appreciate your instructions.

Comment 2

 in the table 2, Would it be possible to standardize the parameters assessed for FVC-related endpoint?

Response

 Thank you for your suggestion. However, we are sorry, but we are unable to standardize the parameters assessed for FVC-related endpoint. The focuSSced study evaluates both FVC and ppFVC. The SENSCIS study evaluates only FVC and the DESIRES study evaluates only ppFVC. Since no single parameter was commonly used in all three clinical trials, it was necessary to use two different parameters (FVC and ppFVC) in Table 3. Note that the focuSSced test emphasizes ppFVC more than FVC, and only ppFVC is described as a key secondary endpoint. Therefore, in Table 3, for the focuSSced study, we have listed values related to ppFVC not FVC. For these reasons, we would appreciate your understanding that we continue to maintain the current descriptions in Table 3.

Comment 3

 a reference for tolicizumab is missing in relation to its mechanism of action line 169 to 171

Response

 Thank you for your valuable feedback. We apologize for not specifying the reference. We have added the necessary references in the text.

Comment 4

 line 332 is it japanese that was written or japan?

Response

 Thank you for your question. We apologize that the description is not clear. It means "Japanese people," so we have changed the term from "Japanese" to "Japanese patients.
